

# Genome-wide characterization of the xyloglucan endotransglucosylase/hydrolase gene family in *Solanum lycopersicum* L. and gene expression analysis in response to arbuscular mycorrhizal symbiosis

Luis G. Sarmiento-López[1], Maury Yanitze López-Espinoza[1], Marco Adán Juárez-Verdayes[2] and Melina López-Meyer[1]

[1] Departamento de Biotecnología Agrícola, Centro Interdisciplinario de Investigación para el Desarrollo Integral Regional Unidad Sinaloa-Instituto Politécnico Nacional, Guasave, Sinaloa, México
[2] Departamento de Ciencias Básicas, Universidad Autónoma Agraria Antonio Narro, Saltillo, Coahuila, México

## ABSTRACT

Xyloglucan endotransglucosylase/hydrolases (*XTHs*) are a glycoside hydrolase protein family involved in the biosynthesis of xyloglucans, with essential roles in the regulation of plant cell wall extensibility. By taking advantage of the whole genome sequence in *Solanum lycopersicum*, 37 *SlXTHs* were identified in the present work. SlXTHs were classified into four subfamilies (ancestral, I/II, III-A, III-B) when aligned to XTHs of other plant species. Gene structure and conserved motifs showed similar compositions in each subfamily. Segmental duplication was the primary mechanism accounting for the expansion of *SlXTH* genes. *In silico* expression analysis showed that *SlXTH* genes exhibited differential expression in several tissues. GO analysis and 3D protein structure indicated that all 37 SlXTHs participate in cell wall biogenesis and xyloglucan metabolism. Promoter analysis revealed that some *SlXTHs* have MeJA- and stress-responsive elements. qRT-PCR expression analysis of nine *SlXTHs* in leaves and roots of mycorrhizal colonized vs. non-colonized plants showed that eight of these genes were differentially expressed in leaves and four in roots, suggesting that *SlXTHs* might play roles in plant defense induced by arbuscular mycorrhiza. Our results provide valuable insight into the function of XTHs in *S. lycopersicum*, in addition to the response of plants to mycorrhizal colonization.

# INTRODUCTION

The plant cell wall is a complex extracellular matrix important to such domains as morphology and growth (*Somerville et al., 2004*). It is composed of cellulose (30%), hemicellulose (30%), pectin (35%), and structural proteins (5%) (*Cosgrove, 2022*). Cellulose and hemicellulose provide rigidity to the wall, whereas pectin provides flexibility and fluidity. Hemicellulose is formed by monosaccharides such as mannan, xylan, and

Corresponding author
Melina López-Meyer, mlopez@ipn.mx

glucomannan linked to a xyloglucan backbone (*Scheller & Ulvskov, 2010*; *Pauly & Keegstra, 2016*; *Voiniciuc, 2022*).

A family of polysaccharides, xyloglucans, are one of the most abundant components in the hemicellulose of monocotyledonous and dicotyledonous plants. Xyloglucans are bonded to adjacent cellulose microfibril surfaces, forming a network that may limit cell wall extensibility while causing loosening when they degrade (*Pauly & Keegstra, 2016*). In addition, xyloglucans play essential roles in controlling cell enlargement, regulating their biosynthesis and metabolism, and functioning as a storage reserve in the seeds of many plant families such as Asteraceae, Brassicaceae, Fabaceae and Solanaceae, where they are accumulated in large quantities to provide energy for the seedling (*Dos Santos et al., 2004*; *Hoch, 2007*).

Xyloglucan endotransglucosylase/hydrolases (XTHs) form a crucial family of xyloglucan-modifying enzymes mainly responsible for the cleavage and rearrangement of xyloglucan backbones in plants (*Hayashi & Kaida, 2011*; *Pauly & Keegstra, 2016*). XTHs are classified within glycoside hydrolase family 16 (GH16; CAZy database; http://www.cazy.org/), whose members have two catalytic activities. Specifically, they can act as an endotransglucosylase (XTE) to catalyze xyloglucan transfer to another xyloglucan molecule, resulting in the elongation of xyloglucan; and as a hydrolase (XEH) that hydrolyzes one xyloglucan molecule, resulting in irreversible xyloglucan chain shortening (*Rose et al., 2002*; *Miedes & Lorences, 2009*; *Behar, Graham & Brumer, 2018*). Many XTHs present both catalytic activities and are important in regulating cell wall extensibility, root elongation, hypocotyl growth, and flower opening (*Dos Santos et al., 2004*; *Wu et al., 2005*; *Harada et al., 2011*).

*XTH* gene family members are highly involved in the regulation of cell wall responses to biotic and abiotic stresses that consequently affect plant growth (*Rose et al., 2002*; *Albert et al., 2004*; *Cho et al., 2006*; *Yan et al., 2019*; *Niraula et al., 2021*). Furthermore, several studies have shown that the expression of *XTH* genes is regulated by plant hormones (*Xu et al., 1996*; *Yokoyama & Nishitani, 2001*; *Jan et al., 2004*; *Osato, Yokoyama & Nishitani, 2006*; *Zhu et al., 2013*; *Han et al., 2016*). Other studies have reported that *XTHs* from *Fragaria chiloensis* are involved in fruit ripening, including in apples and tomatoes (*Miedes & Lorences, 2009*; *Opazo et al., 2010*; *Muñoz Bertomeu, Miedes & Lorences, 2013*; *Méndez-Yañez et al., 2017*).

Different numbers of *XTH* genes have been identified and characterized in plant species such as *Arabidopsis thaliana* (33 genes; *Yokoyama & Nishitani, 2001*), *Oryza sativa* (29 genes; *Yokoyama, Rose & Nishitani, 2004*), *Sorghum bicolor* (35 genes; *Rai et al., 2016*), *Hordeum vulgare* (24 genes; *Fu, Liu & Wu, 2019*), *Actinidia deliciosa* (14 genes; *Atkinson et al., 2009*), *Malus domestica* (11 genes; *Atkinson et al., 2009*), *Glycine max* (61 genes; *Song et al., 2018*), *Solanum lycopersicum* (25 genes; *Saladié et al., 2006*), *Ananas comosus* (48 genes; *Li et al., 2019*), *Brassica rapa* (53 genes; *Wu et al., 2020*), *Brassica oleracea* (38 genes; *Wu et al., 2020*), *Nicotiana tabacum* (56 genes; *Wang et al., 2018*), *Vitis vinifera* L. (34 genes; *Qiao et al., 2022*), *Arachis hypogaea* L. (58 genes; *Zhu et al., 2022*) and *Schima superba* (34 genes; *Yang, Zhang & Zhou, 2022*).

Arbuscular mycorrhizal (AM) symbiosis is a mutualistic interaction between AM fungi from the Glomeromycota phylum and most land plants (*Spatafora et al., 2016*). This
symbiosis improves plant growth, photosynthesis, and nutrient uptake (mainly P) and reduces susceptibility to pathogens in a systemic manner (*Smith & Read, 2008*; *Miozzi et al., 2019*; *Sanmartín et al., 2021*). Mycorrhizal colonization induces a priming state, so plants respond faster and more robustly to pathogen attack (*Pozo & Azcón-Aguilar, 2007*). Since the ectopic expression of defense genes involved in cell wall synthesis can confer resistance to bacteria, fungi, viruses, nematodes, and insects (*Zhang et al., 2019*), it can be hypothesized that some *XTHs* could have a role in the priming mechanism, not only locally in colonized roots but also systemically in shoots. A previous microarray transcriptomic analysis in *M. truncatula* revealed that an *XTH* gene was induced explicitly in shoots of AM plants. After infection with the pathogen *Xanthomonas campestris* showed increased in resistance compared to non-colonized plants (*Liu et al., 2007*). These data are in agreement with an RNA-seq analysis in which cell wall biogenesis-related genes, including some *XTHs*, were differentially regulated in leaves of AM tomato plants in parallel with an increase in resistance against the shoot pathogen *Sclerotinia sclerotiorum* (*Cervantes-Gámez et al., 2016*; *Mendoza-Soto et al., 2022*). This supports the idea that cell wall modification genes, including *XTHs*, play an essential role in shoots of AM-colonized plants to trigger a priming mechanism that improves defense against subsequent pathogen attacks.

Although studies on the identification and characterization of *XTHs* in *S. lycopersicum* are scarce, there are reports on the involvement of some of these proteins in the tomato fruit development (*Saladié et al., 2006*; *Miedes & Lorences, 2009*). The availability of the complete tomato genome sequence provides an opportunity to carry out a comparative analysis of the whole *XTH* gene family. In the present study, we identified all potential *XTH* genes encoded in the *S. lycopersicum* genome. Furthermore, we conducted a bioinformatics analysis to classify *SlXTH* genes by the presence of characteristic motifs, exon-intron organization, chromosomal distribution, and gene duplication events. Finally, the expression patterns of several *SlXTH* genes were characterized by qRT-PCR in shoots and roots of AM tomatoes to investigate the biological importance of this gene family, particularly the response of some of its members in mycorrhiza-colonized plants.

## MATERIALS & METHODS

### Identification of *XTH* family members in *Solanum lycopersicum*

All gene and protein sequence information was retrieved by searching the Phytozome v13 database (http://www.phytozome.net) and the Solanaceae crops genome database (https://solgenomics.net/). To identify all SlXTH proteins, the BLASTP algorithm using the SlXTH14 amino acid sequence was employed to search all potential XTH proteins in the *Solanum lycopersicum* genome. SlXTH14 was selected based on transcriptomic analysis as previously reported by *Cervantes-Gámez et al. (2016)*, in which several cell wall biogenesis-related genes were differentially expressed in tomato leaves in response to AM symbiosis. The Hidden Markov Model (HMM, https://www.ebi.ac.uk/Tools/hmmer/) was used to search the profiles of the SlXTH protein domains PF00722 and PF06955, as previously reported by *Wang et al. (2018)*. The online program SMART (http://smart.embl-heidelberg.de/) was used to identify the conserved domain of candidate SlXTHs, and only

the proteins containing both domains PF00722 and PF06955 were kept for further analysis. The chromosome coordinates of each *SlXTH* genomic sequence (File S1), as well as their coding (File S2), transcript (File S3), and protein (File S3) sequences were obtained from the Phytozome v13 database. Physicochemical parameters for each protein, including predicted molecular weight and isoelectric point (PI), were obtained using tools available at the ExPASy bioinformatics resource portal (https://www.expasy.org/). The subcellular localizations were predicted with ProtComp 9.0 (http://linux1.softberry.com). The SignalP 5.0 server (https://services.healthtech.dtu.dk/service.php?SignalP-5.0) was used to predict the presence of signal peptides. Finally, *SlXTH* genes were nominated as previously reported (*Saladié et al., 2006*), and new sequences were named according to the following numbers.

## Gene structure and motif analysis

Genomic and complete coding DNA (CDS) corresponding to each identified *SlXTH* gene were analyzed for exon-intron distribution. The Gene Structure Display Server (GSDS 2.0) (http://gsds.gao-lab.org/) was employed to obtain graphical representation of the exon-intron organization by comparing the CDS sequences of the *SlXTH* genes to the corresponding genomic DNA sequences (*Hu et al., 2015*). Protein structural motif analysis was performed using the MEME program (https://meme-suite.org/meme/) to predict conserved motifs (10 maximum motifs) in SlXTH proteins as previously reported (*Bailey et al., 2009*). The consensus sequence was analyzed to identify the conserved catalytic motif (DEIDFEFLG) of SlXTH proteins, and the web logo was illustrated using the MEME tool.

## Structurally based sequence alignment and structural prediction of SlXTH proteins

The bioinformatics online tool ESPript (https://espript.ibcp.fr/ESPript/ESPript/) was used to predict the secondary structures in the SlXTH protein sequences and the secondary elements. SlXTH sequences were aligned using ClustalW with default settings to identify shared structural features of SlXTHs, and the PDB databank (https://www.rcsb.org/) was used to locate the XTH crystal protein structure (PDB id: 2UWA; PDB id:1UN1) as previously reported (*Johansson et al., 2004*; *Baumann et al., 2007*). Three-dimensional (3D) structures predicting models of SlXTH proteins were constructed based on the oligomeric state, the maximized percentage identity, ligands, the model quality estimation (QMEAN) and the global quality estimation score (GMQE), using the SWISS-MODEL template library (https://swissmodel.expasy.org/) (*Biasini et al., 2014*).

## *In silico* chromosomal mapping, gene duplication and Ka/Ks estimation

The chromosomal location of each *SlXTH* was obtained from the Phytozome v13 database. The physical location and relative distances of *SlXTH* genes were schematically represented on their respective tomato chromosome using the online server MG2C (http://mg2c.iask.in/mg2c_v2.0/). To analyze gene duplication events, tandem and segmental duplications were considered. A gene pair on the same chromosome located five or fewer gene loci apart and showing more than 90% sequence similarity was considered a tandem duplication, whereas sister gene pairs located on different chromosomes

were considered segmental duplication events. To estimate the selective pressure and divergence time of *SlXTH* genes, amino acid and coding sequences from segmental gene pair duplications were analyzed using the Toolkit for Biologists Tools (TBtools) software (https://github.com/CJ-Chen/TBtools) to determine the Ka (non-synonymous), Ks (synonymous), and Ka/Ks ratio parameters (*Chen et al., 2020*). The approximate time (T) duplication event was estimated using the $(T) = Ks/2\lambda \times 10^{-6}$ million years ago (Mya) for each gene pair, where $\lambda = 1.5 \times 10^{-8}$ substitutions per site per year for dicot plants (*Koch, Haubold & Mitchell-Olds, 2000*).

## Gene ontology (GO) annotation

Gene ontology annotation analysis of *SlXTH* genes was conducted using the Blast2GO software (https://www.blast2go.com/) (*Conesa & Götz, 2008*). Amino acid sequences of each *SlXTH* gene were uploaded to the program and the biological process (BP), cellular compartments (CC), and molecular functions (MF) were determined. In addition, a Blast2GO analysis was performed to BLASTp search, InterPro Scan, mapping and annotation with default settings.

## Analysis of *cis*-acting regulatory elements from the *SlXTH* genes

To predict the *cis*-acting elements in the promoter of *SlXTH* genes, 2.0 kb upstream of the initiation codon (ATG) of each *SlXTH* gene were extracted from the Phytozome v13 database. The upstream sequences were submitted to the online PlantCare (http://bioinformatics.psb.ugent.be/webtools/plantcare/html/) database for the prediction (*Lescot, 2002*) and visualized using GraphPad Prism 6 software.

## Phylogenetic analysis of SlXTH proteins

For insight into the evolutionary relationship among different XTH gene family members, we performed a multiple sequence alignment of the full-length XTH protein sequences from other solanaceous plants such as *N. tabacum*, *S. tuberosum*, *Petunia axillaris* and the model plant *A. thaliana* using ClustalW with default parameters. We analyzed the results with MEGA X (http://www.megasoftware.net) (*Kumar et al., 2018*). The phylogenetic tree was constructed based on the neighbor-joining algorithm with 1000 bootstrap replications, and was visualized using the iTOL online tool (https://itol.embl.de/) (*Letunic & Bork, 2016*).

## Gene expression analysis

RNA-seq expression data for *S. lycopersicum* tissues including leaves, roots, buds, and flowers were downloaded from the GEO database at NCBI (http://www.ncbi.nlm. nih. gov/geo/) and the Solanaceae crops genome database (SRA049915: accession numbers SRX118613, SRX118614, SRX118615, SRX118616) (FDR less than 3%; *q*-value threshold <0.03) (The Tomato Genome Consortium, 2012). The *XTH* expression data were estimated using the expressed 'reads per kilobase of exon per million fragments mapped' (RPKM) value. RPKM values for different tissue were subjected to hierarchical clustering analysis with TBtools software (https://github.com/CJ-Chen/TBtools). Finally, the data were normalized to examine differences in the expression of the same gene in different samples and represented as a heatmap with TBtools.

## Plant material and growth conditions

The arbuscular mycorrhizal fungi *Rhizophagus irregularis* was provided by the CIIDIR-SINALOA at the Instituto Politécnico Nacional in Sinaloa, Mexico. The inoculum was grown according to previously reported methods (*Bécard & Fortin, 1988*).

*S. lycopersicum* (var. Missouri) seeds were surface-sterilized. Tomato seeds were planted in germination trays with a mixture of sterilized vermiculite and sand (3:1 v/v) and maintained at 25 °C. Four-week-old tomato plants were transplanted individually to pots (1 L) with the same substrate. At this time, tomato plants were inoculated with 500 spores of *R. irregularis* (M+ treatment). AM spores were prepared from an axenic carrot root culture colonized with *R. irregularis* and extracted as previously described by *Cervantes-Gámez et al. (2016)*. Control samples consisted of mock-inoculated plants (*i.e.,* non-colonized, M-treatment) with the last rinse of the spore inoculum wash. All plants were watered once per week with distilled water and twice per week with 30 mL of half-strength Hoagland nutrient solution with 50 μM $KH_2PO_4$ as the final phosphate concentration to favor the mycorrhizal colonization (*Hoagland & Arnon, 1950*). One-half of the root system and whole leaves from each M+ and M- tomato plant were harvested four weeks after *R. irregularis* inoculation. The collected plant material was immediately frozen in liquid nitrogen and stored at −80 °C for subsequent RNA extraction. Five biological replicates were employed per treatment and two independent experiments were performed.

The other half of each plant root system was fixed in 50% ethanol, clarified in 20% KOH, neutralized in 0.1 M HCl, and stained in 0.05% trypan blue in lactoglycerol (*Phillips & Hayman, 1970*). Roots were maintained in lactoglycerol 1:1:1 (water/lactic acid/glycerol) and observed by light microscopy (BOECO Germany, BM-180). Mycorrhizal colonization was confirmed as previously reported by *Mendoza-Soto et al. (2022)*.

## RNA extraction, primer design, and qRT-PCR analysis

Nine out of the 37 identified *SlXTH* genes were selected to experimentally determine their expression in leaves and root tissues of mycorrhizal colonized and non-colonized plants based on the occurrence of defense-related regulatory elements within their promoter sequences.

Total RNA was isolated from leaves and roots of non-colonized (M-) and colonized (M+) plants using TRIzol reagent (Invitrogen, Carlsbad, CA), following the manufacturer's protocol. The complementary DNA synthesis was performed as previously reported (*Cervantes-Gámez et al., 2016*). The 3′ untranslated regions (UTR) of each gene were used to design qPCR primers for gene specificity. The primers used are listed in Table S1. Melting temperature (Tm) and GC content were calculated using Oligo Calc (http://biotools.nubic.northwestern.edu/OligoCalc.html). qRT-PCR was performed using SYBR Green (QIAGEN, USA) and quantified on a Rotor-Gene Q (QIAGEN, USA) real-time PCR thermal cycler. qRT-PCR was programmed for 40 cycles, denaturing at 95 °C for 15 s, annealing at 58 °C for 30 s, and extension at 72 °C for 30 s. Amplification of a single PCR product was verified by thermal gradient PCR and melting curve qRT-PCR analysis. The elongation factor 1-α (*SlEF1- α*) gene was used for normalization. The relative expression of *SlXTH* genes was calculated by the $2^{-\Delta CT}$ method (*Livak & Schmittgen,*
*2001*). Five biological replicates for each condition (non-colonized and colonized plants) were evaluated, and two independent experiments were performed with similar results. Data from one of the experiments are shown.

### Data analysis

For the relative expression of each *SlXTH* gene, the paired Student's *t*-test was used to evaluate the significance of differences between non-colonized (M-) and colonized (M+) tomato plants. All data were checked for normal distributions (Shapiro–Wilk's test) before statistical analyses, which were performed using the scientific data analysis and graphing software SigmaPlot for Windows, version 11.0.

## RESULTS

### Identification and characterization of the *SlXTH* gene family

A comprehensive genome-wide screening of the tomato database was executed to identify all *SlXTH* genes. As a result, 37 *SlXTH* genes were identified, including some novel members of the family. All *SlXTH* genes identified within the tomato genome showed the conserved PF00722 (glycosyl hydrolases family 16) and PF06955 (xyloglucan endotransglucosylase C-terminus) domains by using Pfam analysis (Fig. S1), which verifies and validates the sequence search results. The 37 *SlXTH* genes were named *SlXTH1* to *SlXTH37* based on a previously reported work in which *SlXTH1* to *SlXTH25* were already identified in *S. lycopersicum* (*Saladié et al., 2006*). The characteristics of each sequence, including gene ID number and length of the genomic, transcript, coding DNA (CDS) and amino acid sequences, as well as molecular weight (MW), isoelectric point (PI), and chromosome coordinates, are summarized in Table 1 and Table S2. The length of SlXTH proteins ranged from 274 (SlXTH31) to 372 (SlXTH26) amino acids, with the predicted CDS ranging from 825 to 1,119 bp and the calculated MW varying between 0.69 and 1.63 kDa. SlXTH26 was the most significant XTH protein (Table 1). The theoretical PI values of SlXTH ranged from 4.85 (SlXTH27) to 9.51 (SlXTH14) due to the differences in ionic strength and pH in the amino acids present in these proteins (Table 1).

Subcellular localization prediction revealed that most of the SlXTH proteins (32 out of 37 SlXTHs) were located on the plasma membrane. In contrast, SlXTH5, SlXTH6, SlXTH14, SlXTH26, and SlXTH36 were predicted to localize in the extracellular region (Table S2). In addition, the signal peptide prediction indicated that all SlXTH proteins contain signal peptide sequences exceptSlXTH13, SlXTH18 and SlXTH22 (Table S2).

### Gene structure and conserved motif analysis of SlXTH proteins

To investigate the structural diversity of the *SlXTH* genes, exon and intron structures of the 37 *SlXTH* genes were determined by aligning their CDS and genomic sequences using the GSDS server. We also constructed a phylogenetic tree using full-length deduced amino acid sequences of the *SlXTH* genes, presented with the exon and intron distribution in Fig. 1. The phylogenetic tree shows that the *SlXTH* genes are divided into two major subfamilies: subfamily I/II and III. Subfamily I/II has 30 gene members, while subfamily III has seven. Structural analysis of *SlXTH* genes showed that each subfamily's most closely related genes

**Table 1  Structural features of Xyloglucan Endotransglucosylase/Hydrolase (*XTH*) family genes in tomato (*Solanum lycopersicum* L).**

| Name | SGN I.D. | Size | | | | | | | |
|---|---|---|---|---|---|---|---|---|---|
| | | Genomic | Transcript | CDS | 5′ UTR | 3′ UTR | Protein (aa) | PI | MW |
| *SlXTH1* | Solyc01g099630.2.1 | 2728 | 1315 | 891 | 98 | 326 | 296 | 8.80 | 0.86 |
| *SlXTH2* | Solyc07g009380.2.1 | 2322 | 1264 | 828 | 62 | 374 | 275 | 8.54 | 1.02 |
| *SlXTH3* | Solyc03g093130.2.1 | 1249 | 1072 | 864 | 40 | 168 | 287 | 5.23 | 0.69 |
| *SlXTH4* | Solyc11g065600.1.1 | 2343 | 882 | 882 | – | – | 293 | 7.64 | 0.86 |
| *SlXTH5* | Solyc01g081060.2.1 | 3233 | 1366 | 1014 | 77 | 275 | 337 | 7.20 | 1.00 |
| *SlXTH6* | Solyc11g066270.1.1 | 2105 | 891 | 891 | – | – | 296 | 8.51 | 1.12 |
| *SlXTH7* | Solyc02g091920.2.1 | 1632 | 1165 | 888 | 38 | 239 | 295 | 7.60 | 1.16 |
| *SlXTH8* | Solyc04g008210.1.1 | 2421 | 993 | 993 | – | – | 330 | 6.07 | 0.85 |
| *SlXTH9* | Solyc12g011030.1.1 | 1106 | 834 | 834 | – | – | 277 | 9.10 | 1.18 |
| *SlXTH10* | Solyc07g056000.2.1 | 1276 | 1098 | 864 | 58 | 176 | 287 | 8.29 | 0.82 |
| *SLXTH11* | Solyc12g017240.1.1 | 1080 | 870 | 870 | – | – | 289 | 8.53 | 0.82 |
| *SlXTH12* | Solyc09g092520.2.1 | 2075 | 1084 | 831 | 60 | 193 | 276 | 9.08 | 1.02 |
| *SlXTH13* | Solyc07g006850.1.1 | 4272 | 1029 | 1029 | – | – | 342 | 8.88 | 1.08 |
| *SlXTH14* | Solyc09g008320.2.1 | 2771 | 1255 | 897 | 84 | 275 | 298 | 9.51 | 1.12 |
| *SlXTH15* | Solyc03g031800.2.1 | 2179 | 1085 | 888 | 78 | 120 | 295 | 4.94 | 0.86 |
| *SlXTH16* | Solyc07g052980.2.1 | 1949 | 1125 | 891 | 54 | 158 | 296 | 5.94 | 1.16 |
| *SlXTH17* | Solyc07g055990.2.1 | 1968 | 1108 | 873 | 51 | 184 | 290 | 8.18 | 1.16 |
| *SlXTH18* | Solyc12g007260.1.1 | 2066 | 879 | 879 | – | – | 292 | 9.21 | 0.84 |
| *SlXTH19* | Solyc05g046290.2.1 | 2715 | 995 | 867 | – | 128 | 288 | 5.92 | 0.94 |
| *SlXTH20* | Solyc07g006870.2.1 | 1898 | 1074 | 849 | 42 | 183 | 282 | 5.28 | 1.03 |
| *SlXTH21* | Solyc01g005120.2.1 | 3224 | 1497 | 1008 | 216 | 273 | 335 | 6.86 | 0.85 |
| *SlXTH22* | Solyc12g007270.1.1 | 2341 | 882 | 882 | – | – | 293 | 5.42 | 0.99 |
| *SlXTH23* | Solyc02g080160.2.1 | 1909 | 1179 | 897 | 43 | 239 | 298 | 6.38 | 0.93 |
| *SlXTH24* | Solyc03g093120.2.1 | 1243 | 1067 | 861 | 40 | 166 | 286 | 5.23 | 0.69 |
| *SlXTH25* | Solyc05g005680.2.1 | 1948 | 1323 | 942 | 73 | 308 | 313 | 6.53 | 1.23 |
| *SlXTH26* | Solyc08g076080.2.1 | 4195 | 1371 | 1119 | – | 252 | 372 | 9.25 | 1.00 |
| *SlXTH27* | Solyc10g005350.2.1 | 1997 | 1012 | 867 | – | 145 | 288 | 4.85 | 0.86 |
| *SlXTH28* | Solyc03g098430.2.1 | 1477 | 1092 | 852 | 79 | 161 | 283 | 5.82 | 1.16 |
| *SlXTH29* | Solyc12g007250.1.1 | 1413 | 876 | 876 | – | – | 291 | 8.83 | 0.84 |
| *SlXTH30* | Solyc05g053700.1.1 | 2956 | 849 | 849 | – | – | 282 | 8.16 | 1.26 |
| *SlXTH31* | Solyc06g083400.1.1 | 2157 | 825 | 825 | – | – | 274 | 5.10 | 1.52 |
| *SlXTH32* | Solyc07g006860.2.1 | 1832 | 1168 | 858 | 76 | 234 | 285 | 7.58 | 0.99 |
| *SlXTH33* | Solyc11g040140.1.1 | 2866 | 903 | 903 | – | – | 300 | 8.61 | 1.01 |
| *SlXTH34* | Solyc01g106650.2.1 | 1422 | 1044 | 882 | 58 | 104 | 293 | 8.19 | 1.28 |
| *SlXTH35* | Solyc03g093080.2.1 | 1216 | 1035 | 861 | 49 | 125 | 286 | 5.39 | 0.70 |
| *SlXTH36* | Solyc11g017450.1.1 | 1135 | 960 | 960 | – | – | 319 | 5.04 | 1.63 |
| *SlXTH37* | Solyc03g093110.2.1 | 1262 | 1082 | 864 | 52 | 166 | 287 | 5.40 | 0.69 |

**Notes.**

SGN I.D., Solanaceae Genome Network identification; CDS, coding DNA sequence; UTR, untranslated region; aa, amino acids; PI, isoelectric point; MW, Molecular Weight.

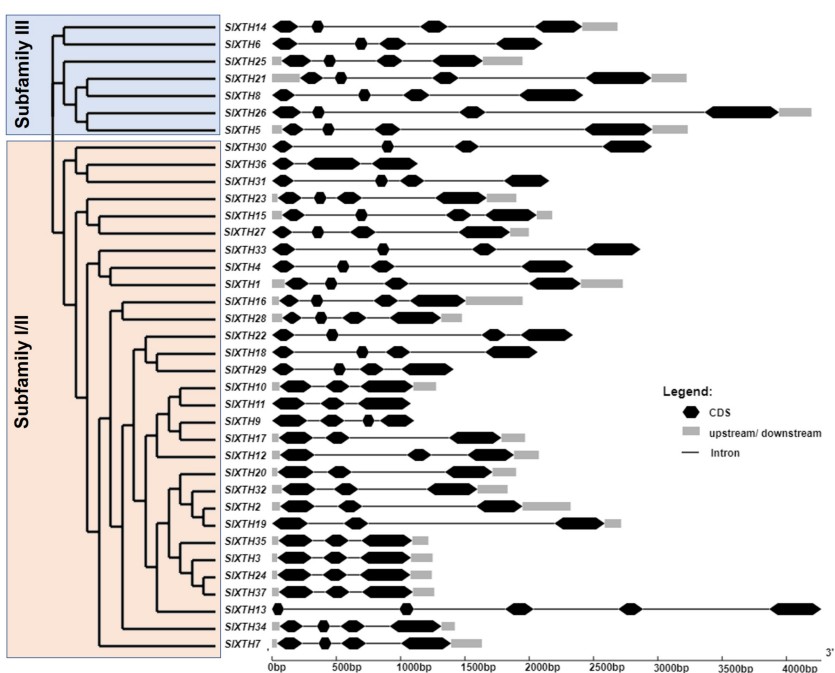

**Figure 1** **Analysis of phylogenetic relationships and gene structure of the *Sl XTH* gene family.** A phylogenetic tree of the *SlXTH* gene family was constructed using the neighbor joining method, and then classified into two subfamilies. Subfamily I/II is in pink and subfamily III is in blue.

share similar exon and intron numbers. For example, members from subfamily I/II mostly contain three introns and four exons distributions, except*SlXTH13*, which presents five exons in its coding region. All members of subfamily III also contain three introns and four exons in their coding region. On the other hand, the presence or absence of 5′ and 3′ UTR was not exclusively associated with either of the two subfamilies. For example, *SlXTH6* and *SlXTH8* from subfamily III, and *SlXTH30*, *SlXTH31*, *SlXTH33*, *SlXTH22*, *SlXTH18*, *SlXTH29*, *SlXTH11*, *SlXTH9*, and *SlXTH13* from subfamily I/II do not have any 5′ or 3′ UTRs (Fig. 1).

To further characterize the SlXTH family, MEME motif detection software was used to predict potentially conserved motifs. A total of ten conserved motifs with lengths of ten amino acids were identified. Motif compositions differed in members from the two subfamilies (Fig. 1 and Fig. S2). For example, all SlXTH members from subfamilies I/II and III showed the presence of the ten highly conserved motifs, except for SlXTH36 and SlXTH31 from subfamily I/II, which only have seven and six conserved motifs, respectively (Fig. S2A). Furthermore, multiple sequence alignment of all SlXTH proteins revealed the conserved amino acid motif DEIDFEFLG, which is responsible for the catalytic activity as well as being the most characteristic motif of this family (Fig. S2B), indicating that this conserved core motif is an essential for XTH proteins, and suggesting that all of these proteins have a similar function. The majority of the SlXTH proteins within the same

subfamily showed identical gene structure and motif compositions, consistent with the phylogenetic analysis of the whole *XTH* gene family.

## Structural prediction of SlXTH proteins

The alignments of SlXTHs with a xyloglucan endotransglycosylase crystal protein structure (PDB id: 2UWA and PDB id:1UN1) were used to predict the secondary structures of the SlXTH proteins with ESPript (Figs. S3 and S4). All SlXTH protein members in subfamily I/II and subfamily III had similar structures to the reference crystal protein structure. Twenty-eight subfamily I/II members showed a conserved position of the N-glycosylation site at amino acid 99. However, this was not found in two SlXTH proteins (SlXTH31 and SlXTH36) (Fig. S3, see label *). Amino acid 116 was also conserved in all members of subfamily III (Fig. S4, see label *). The active site (ExDxE) containing the residues responsible for catalytic activity was highly conserved in all SlXTH family members (Fig. S3 and S4, see label AS). In addition, all members possess the XET/XEH C-terminal extension, a characteristic fingerprint among XTHs from other plant species.

A tertiary (3D) protein model of SlXTHs might help to understand XTH enzyme's structure and possible mode of action (Table S3). Most SlXTHs from the same subfamily (I/II and III) showed similar 3D structures with percentage identities between 36.54 and 77.32%, indicating a reliable structure prediction. In addition, essential ligands were predicted based on their chemical identity. For example, ligands for $\beta$-D-glucopyranose, $\alpha$-D-xylopyranose and $\beta$-D-galactopyranose were identified in 34 SlXTH proteins but not in SlXTH8, SlXTH14, and SlXTH21, in which no ligands sites were detected (Table S3).

## Chromosome mapping and gene duplication analysis of *SlXTH* genes

The chromosome coordinates of all *SlXTH* genes were obtained from the Phytozome v13 database (Table S1), and their chromosomal locations were mapped using the online server MG2C (http://mg2c.iask.in/mg2c_v2.0/) (Fig. 2). *SlXTH* genes were heterogeneously distributed among all chromosomes across the tomato genome. The most significant number of *SlXTH* genes were located on chromosomes 12, 3, and 7, with five, six, and seven *SlXTH* genes, respectively. In contrast, chromosomes 4, 6, 8, and 10 had only one *SlXTH* gene. The other chromosomes contained between two and four *SlXTH* genes (Fig. 2). Finally, no *SlXTH* gene was found on chromosome 0.

Tandem and segmental duplications reveal information about the expansion of new gene family members and evolutionary functions in plants (*Ganko, Meyers & Vision, 2007*). Tandem duplications during *SlXTH* evolution were investigated using the Smith-Waterman algorithm alignment. Two *SlXTH* gene pairs (*SlXTH3/SlXTH37* and *SlXTH24/SlXTH35*) were confirmed to be tandem duplicated since sequence similarity was higher than 90% (Table S4). Both *SlXTH* gene pairs are on chromosome 3 (Fig. 2, see label *). A total of 12 segmental duplication events were identified based on phylogenetic analysis, which includes nine sister pairs (*SlXTH36/SlXTH3, SlXTH15/SlXTH27, SlXTH4/SlXTH1, SlXTH16/SlXTH28, SlXTH18/SlXTH29, SlXTH10/SlXTH11, SlXTH9/SlXTH17, SlXTH2/SlXTH19, SlXTH24/SlXTH37*) from subfamily I/II, and three sister pairs (*SlXTH14/SlXTH6, SlXTH21/SlXTH8, SlXTH26/SlXTH5*) from subfamily III

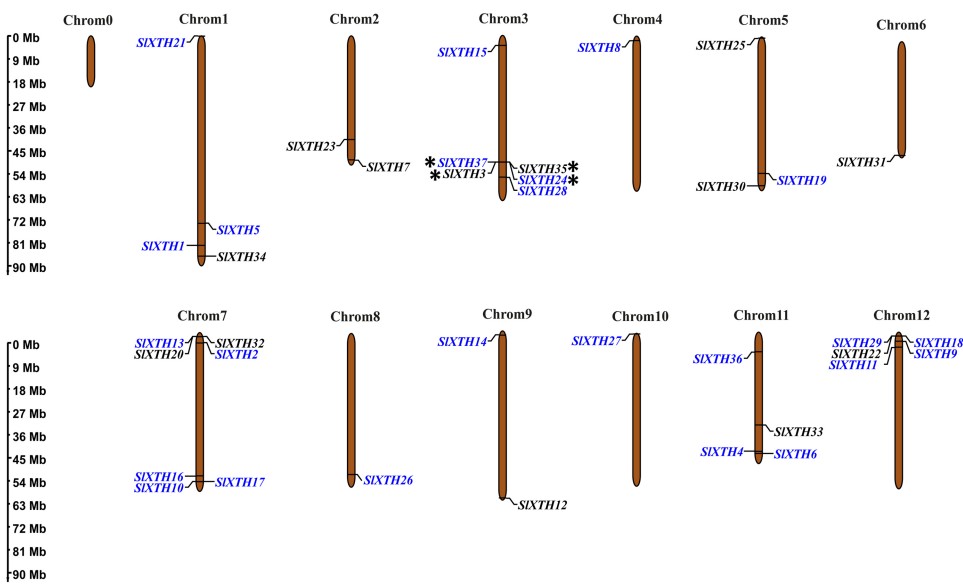

**Figure 2** **Distribution of Sl*XTH* gene members on *S. lycopersicum* chromosomes.** Asterisks indicate tandem duplications, and names in blue indicate segmental duplications. The chromosome scale is in millions of bases (Mb).

(Fig. 1 and 2, see names in blue). The Ka/Ks parameters were evaluated to determine the divergence after duplication. Interestingly, 10 of the 12 sister pairs had Ka/Ks <0.5, which indicates purification selection during evolution. Furthermore, divergence times were estimated to have occurred between 7.4 and 233.33 million years ago (Table S5).

## Gene ontology (GO) analysis of *SlXTH* genes

GO analysis was performed on the entire *SlXTH* gene family using Blast2GO software (Fig. S5). *SlXTH* genes are involved in biological processes such as cell wall organization, cell wall biogenesis, and xyloglucan metabolic processes (Fig. S5A). Molecular function and cellular compartment results revealed that all members of the *SlXTH* family were located in the cell wall and apoplastic region. However, some *SlXTH* gene members were found to be integral membrane component (Fig. S5B) and had hydrolase and transferase activities (Fig. S5C). The biological processes, molecular functions, and cellular features of each SlXTH protein are specified in Table S6.

## Phylogenetic analysis of the SlXTH proteins

One hundred fifty-one full-length XTH protein sequences from *S. lycopersicum*, *S. tuberosum*, *P. axillaris*, *N. tabacum* and *A. thaliana* were used to construct a phylogenetic tree based on the neighbor-joining method (Fig. 3; all sequences are provided in File S5). According to this analysis, XTH members are divided into three major subfamilies: an ancestral subfamily (purple branch), subfamily I/II (black branch), and subfamily III, which is divided into subfamilies III-A (pink branch) and III-B (brown branch) (Fig. 3). Nine SlXTHs were clustered in the ancestral subfamily, which includes three SlXTH proteins (SlXTH30, SlXTH31, and SlXTH36). In subfamily III-A, nine XTHs were grouped,

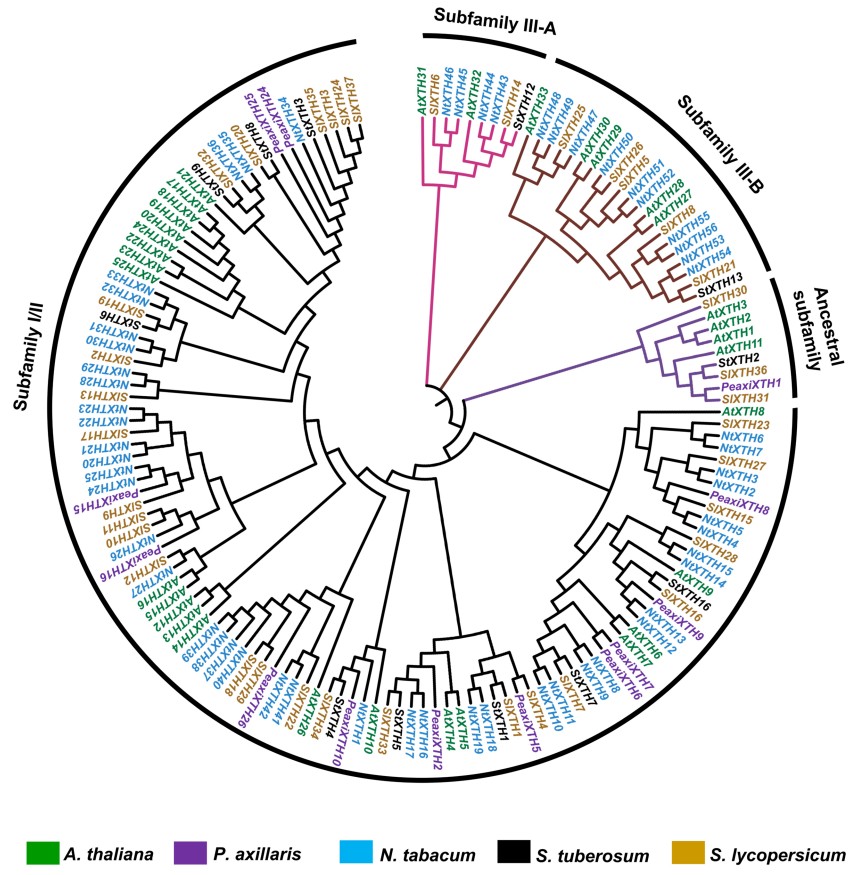

**Figure 3** **Phylogenetic analysis of XTH proteins in different plant species.** XTH proteins corresponding to five different plant species. The tree was constructed using the neighbor joining method with 1,000 bootstrap replicates. The branches correspond to the four phylogenetic subfamilies. Protein names in brown represent *S. lycopersicum*, in blue *N. tabacum*, in purple *P. axillaris*, in black *S. tuberosum*, and in green *A. thaliana*.

two of which were SlXTH proteins. Subfamily III-B contained 21 proteins, six of which were SlXTH proteins. The remaining SlXTHs belonged to subfamily I/II, which includes most of the XTH members from *P. axillaris*, *A. thaliana*, *S. tuberosum*, *N. tabacum*, and *S. lycopersicum* (Fig. 3).

## Analysis of *cis*-acting regulatory elements from the *SlXTH* genes

To further study the potential regulatory elements in the promoters of each member of the *SlXTH* gene family, 2.0 kb of the promoter sequence of each gene was extracted from the tomato genome database and a *cis*-acting regulatory element analysis was conducted (File S6). This analysis revealed that *SlXTH* promoters many regulatory elements, including some involved in cell development, stress-related elements, and hormone regulation (Fig. 4). Methyl jasmonate (MeJa)-responsive regulatory elements were identified in 21 *SlXTH* promoters (Fig. 4, see brown rectangles with a dot). Defense- and stress-responsive cis-elements were found in *SlXTH7*, *SlXTH15*, *SlXTH29*, *SlXTH33*, and *SlXTH36* (Fig. 4, see purple rectangles with a minus sign). Wound-responsive elements were found in the

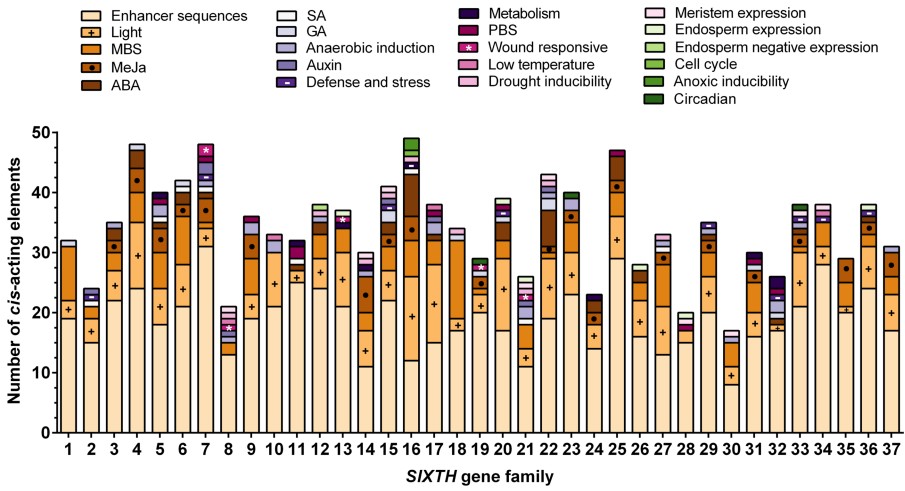

**Figure 4  Putative *cis*-acting regulatory elements in the promoter region of *Sl XTH* genes.** Regulatory elements are represented by different colored boxes and their functions. MYB binding site, MBS; methyl jasmonate, MeJa; abscisic acid, ABA; salicylic acid, SA; gibberellic acid, GA; protein binding site, PBS.

promoter sequences of *SlXTH7*, *SlXTH8*, *SlXTH13*, *SlXTH19*, and *SlXTH21* genes (Fig. 4, see pink rectangles with an asterisk). All 37 *SlXTH* genes contain many light-responsive elements (Fig. 4, see a brownish rectangle with a plus sign). A drought-inducible MYB binding site (MBS) was found in almost all *SlXTH* gene promoters, except *SlXTH10*, *SlXTH11*, *SlXTH24*, *SlXTH28*, and *SlXTH32* (Fig. 4, see the light pink rectangles). The other *SlXTH* members showed different regulatory elements involved in cell development, with roles in meristem expression, endosperm expression, and the cell cycle (Fig. 4). This result indicates that the *SlXTH* gene family members are involved in different biological processes and can respond to various biotic and abiotic stresses.

## Expression profile of *SlXTH* genes in selected tomato tissues

The expression patterns of *SlXTH* genes in different tissues were analyzed using the temporal and spatial expression information from public RNA-seq projects (SGN database) in RPKM values (Fig. 5). Nineteen *SlXTH* genes were expressed in at least one tissue, while 18 were either not expressed in any of the tested tissues or their expression was relatively low (Fig. 5). *SlXTH5*, *SlXTH7*, and *SlXTH16* were highly expressed in leaves, whereas *SlXTH1*, *SlXTH2*, *SlXTH8*, and *SlXTH11* showed less expression in this tissue, and expression was almost undetected in the other members (Fig. 5). *SlXTH16* was highly expressed in leaves, roots, and buds, whereas flowers showed low expression. *SlXTH1* and *SlXTH21* were mainly expressed in flowers, whereas *SlXTH14* was expressed in roots and buds. Some *SlXTH* members, such as *SlXTH6* and *SlXTH9*, were explicitly expressed in roots. Expression in the other *SlXTH* genes was either low or undetected (Fig. 5). No RNA-seq study of AM-colonized shoots in tomatoes is available in the SGN database.

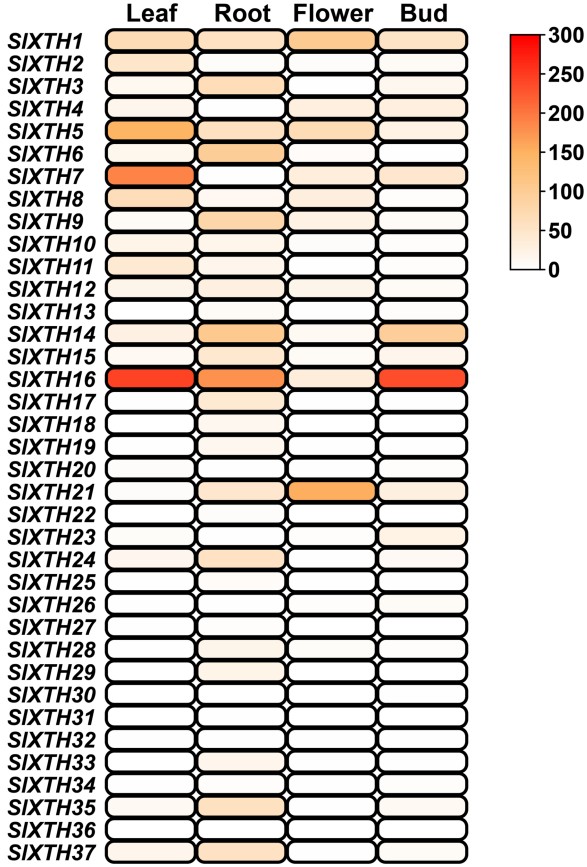

**Figure 5** *SlXTH* **gene expression in different tissues.** Heat map representation of RPKM values for the *SlXTH* genes in tomato vegetative tissues (root, leaf, bud, flower) derived from RNA-seq data (SGN database) for *S. lycopersicum* cv. Heinz. The expression level of *SlXTH* is represented by the color intensity.

## Expression profile of *SlXTH* genes in response to arbuscular mycorrhizal symbiosis

To study the possible role of *SlXTH* family members in the response of tomato plants to AM colonization, we experimentally evaluated the expression levels of some *SlXTH* genes in roots and leaves of colonized (M+) and non-colonized (M-) plants by qRT-PCR. Eight *SlXTH* genes (*SlXTH2*, *SlXTH3*, *SlXTH6*, *SlXTH7*, *SlXTH9*, *SlXTH14*, *SlXTH21*, and *SlXTH35*) were selected based on the fact that they presented at least one cis-regulatory element responsive to defense, stress, wounds, and MeJa, which are all known or postulated to be involved in the modulation of plant defense and priming (Fig. S6). In addition, *SlXTH17*, which does not contain any defense-responsive regulatory elements, was also included in the analysis (Fig. 4 and S6). Microscopy observations of tomato roots revealed that symbiotic structures such as intraradical hyphae, vesicles, and arbuscules were observed in colonized (M+) plants (Fig. 6A, see labels ih, V, and *). As expected, no symbiotic structures were observed in non-colonized (mock, M-) tomato plants (Fig. 6B). In addition, RT-PCR was performed using tomato mycorrhiza-specific phosphate

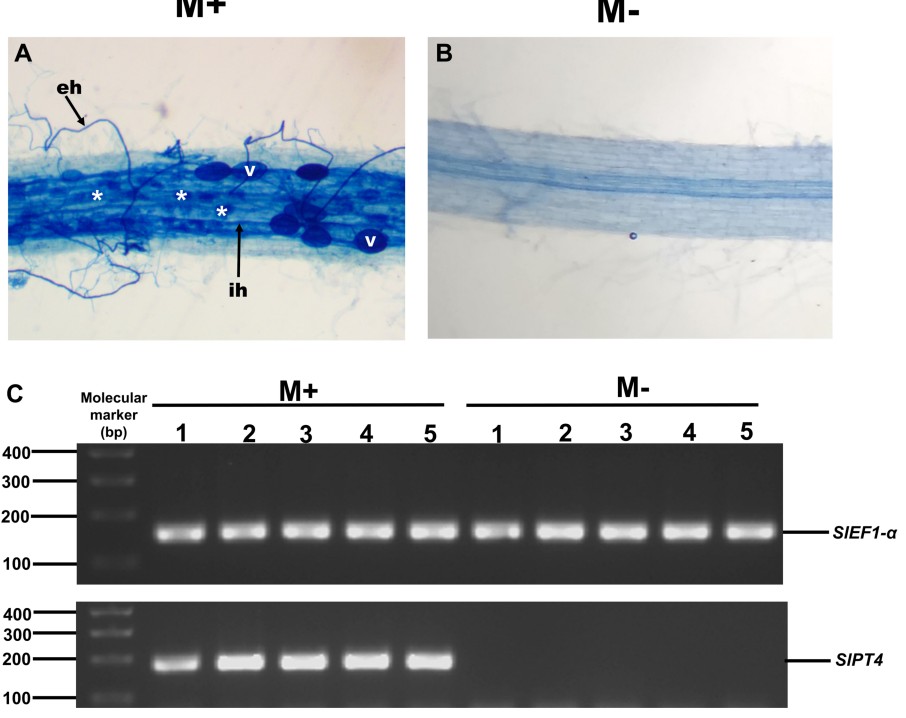

**Figure 6 Mycorrhiza colonization of tomato roots.** Root segments of *S. lycopersicum* colonized (M+; A) and non-colonized (M-; B) plants were analyzed by light microscopy after trypan blue staining. (C) Transcript accumulation of the *SlPT4* gene in roots of *S. lycopersicum* plants. Lanes 1–5 are individual replicates of *R. irregularis* colonized (M+) and non-colonized (M-) plants. Lane MM, molecular size marker. *SlEF1-α* was used as a reference gene. Vesicles, V; intraradical hyphae, ih; extraradical hyphae, eh; arbuscules: *.

transporters (*SlPT4*) as a molecular marker of mycorrhiza colonization in tomato roots, and *SlPT4* transcript accumulation was only detected in the roots of colonized (M+) plants (Fig. 6C).

Differential expression of *SlXTHs* was observed in leaves and roots in response to AM symbiosis. In leaves, only *SlXTH2* showed higher relative expression in M+ plants as compared to M- plants, whereas *SlXTH3*, *SlXTH6*, *SlXTH7*, *SlXTH9*, *SlXTH14*, *SlXTH21* and *SlXTH35* showed downregulation (Fig. 7). The expression of *SlXTH17* was unchanged regardless of the symbiotic status of tomato plants in leaves. In roots, most *SlXTH* genes exhibit no differential change in expression profile in M+ plants compared to M- plants (Fig. 8). Only *SlXTH7* and *SlXTH35* were upregulated in response to AM colonization (M+) compared to the control plants (M-), whereas *SlXTH3* and *SlXTH21* were downregulated. *SlXTH17* was not expressed in tomato roots, regardless of the plant's symbiotic status (Fig. 8). These results indicate that several *SlXTH* genes likely play critical roles in the tomato response to AM symbiosis, suggesting that these genes, which contain regulatory elements involved in plant defense, could participate in the defense priming process and that they are regulated by AM symbiosis.

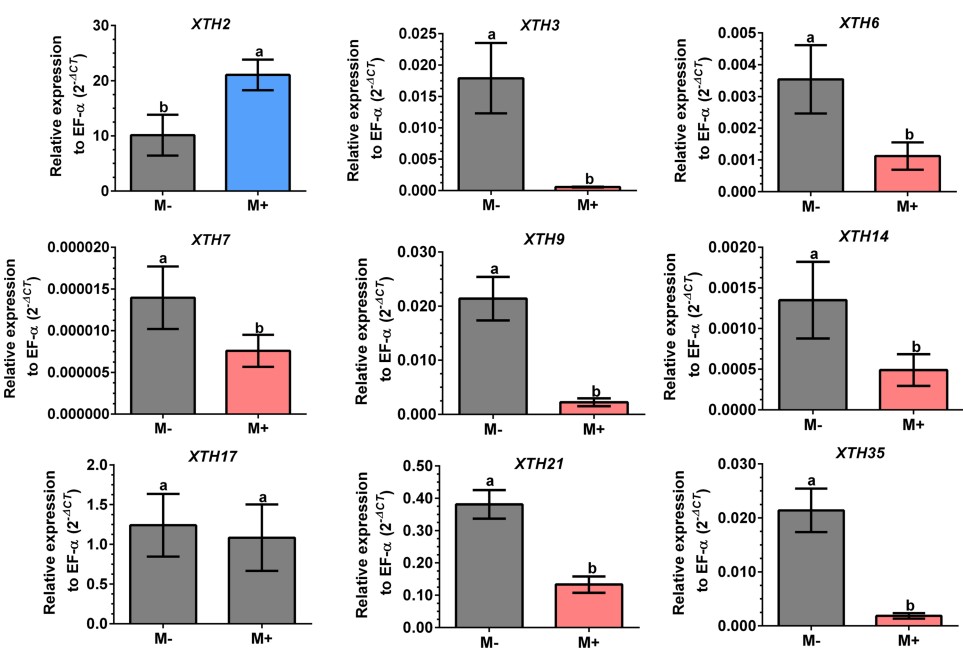

**Figure 7 Differential transcript accumulation of *SlXTH* gene members in tomato leaves in response to AM symbiosis.** Expression levels of *SlXTH* genes in non-colonized (M-) and *R. irregularis* colonized (M+) plants relative to expression of the constitutive gene *SlEF1-α*, calculated using the $2^{-\Delta CT}$ method. Bars represent the mean $\pm$ standard deviation of three biological and three technical replicates. Different letters indicate significant differences according to Student's $t$-test ($p < 0.05$). The specific $p$ value for each *Sl XTH* are represent in Table S7.

## DISCUSSION

Xyloglucan endotransglucosylase/hydrolases (*XTHs*) are a group of xyloglucan modifying-enzymes that have essential roles in the cleavage and rearrangement of the cell wall, affecting its extensibility in plants (*Pauly & Keegstra, 2016*). Twenty-five (25) *XTH* sequences in tomatoes have been reported so far (*Saladié et al., 2006*). Furthermore, based on the release of the tomato genome (*The Tomato Genome Consortium, 2012*), we identified 37 *SlXTH* gene members by genome-wide screening. Consistent with this, large numbers of *XTHs* have been found in other plant species (*Yokoyama & Nishitani, 2001*; *Yokoyama, Rose & Nishitani, 2004*; *Saladié et al., 2006*; *Atkinson et al., 2009*; *Rai et al., 2016*; *Song et al., 2018*; *Wang et al., 2018*; *Fu, Liu & Wu, 2019*; *Li et al., 2019*; *Wu et al., 2020*; *Zhu et al., 2022*; *Qiao et al., 2022*; *Yang, Zhang & Zhou, 2022*).

It is well known that genes structural and physicochemical features are related to their functionality (*Baumann et al., 2007*). In this work, we found differences in gene structure, such as sequence length, exon-intron distribution, molecular weight, and isoelectric point, which suggests that some SlXTH members are functionally different. Furthermore, the 37 SlXTHs described in the present work were divided into two subfamilies, subfamily I/II and subfamily III. Conserved motifs analysis indicates that SlXTHs from subfamily I/II and III have ten conserved motifs, whereas SlXTH31 and SlXTH36 from subfamily I/II have only

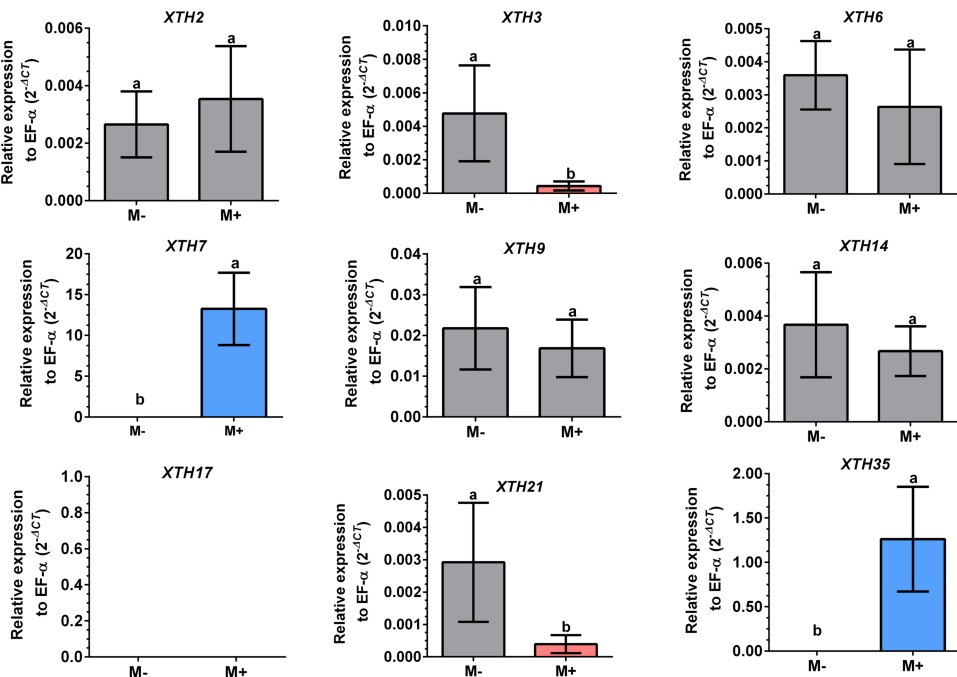

**Figure 8** **Differential transcript accumulation of *SlXTH* gene members in tomato roots in response to AM symbiosis.** Expression levels of *SlXTH* genes in non-colonized (M-) and *R. irregularis* colonized (M+) plants relative to expression of the constitutive gene *SlEF1-α*, calculated using the $2^{-\Delta CT}$ method. Bars represent the mean ± standard deviation of three biological and three technical replicates. Different letters indicate significant differences according to Student's *t*-test ($p < 0.05$). The specific *p* value for each *Sl XTH* are represent in Table S7.

six and seven, respectively. This is consistent with previous reports for other XTHs (*Behar, Graham & Brumer, 2018*; *Wu et al., 2020*; *Qiao et al., 2022*; *Yang, Zhang & Zhou, 2022*).

Despite these differences, all SlXTHs exhibit a highly conserved motif (ExDxE) that acts as the catalytic site for both XET and XEH activity, suggesting that it has been conserved to maintain standard functionality in all members of this family, regardless of any sequence differences among them (*Kaewthai et al., 2013*; *Wang et al., 2018*; *Li et al., 2019*). *Shinohara & Nishitani (2021)* describe that XET and XEH activities are related to extension in loop 2, which is longer than the other subfamilies' loop extension, and the N-glycosylation site, which confers differences in their enzymatic activities between subfamilies.

Previous studies have reported that proteins showing XET activity belong to the most parts of subfamilies I/II, III-B, and the ancestral, while proteins showing a combined function of XET and XEH are included primarily in subfamily III-A (*Rose et al., 2002*; *Baumann et al., 2007*; *Miedes & Lorences, 2009*; *Kaewthai et al., 2013*).

Phylogenetic distribution of XTH proteins from *S. tuberosum*, *N. tabacum*, *P. axillaris*, and *A. thaliana* reveals that the number of genes in subfamily III-A is the smallest, while the number in subfamily I/II is the largest.

In Arabidopsis, two homolog proteins, AtXTH31 and AtXTH32, belong to subfamily III-A, which was confirmed to exhibit XEH activity under *in vitro* conditions (*Kaewthai*

*et al., 2013*). Also, SlXTH6 showed hydrolytic activity (XEH) during fruit growth in *S. lycopersicum* (*Baumann et al., 2007*). According to our phylogenetic results, Arabidopsis proteins AtXTH31 and AtXTH32 and tomato SlXTH6 are grouped in subfamily III-A, which suggests that all members of this subfamily, including SlXTH14, could have the same enzymatic activity. Subfamily III-B includes, notably, the SlXTH5, SlXTH8, and AtXTH27 gene products, previously reported to have XET activity (*Campbell & Braam, 1999*; *Saladié et al., 2006*; *Baumann et al., 2007*). Consistent with this, our results showed that three additional SlXTH (SlXTH21, SlXTH25, and SlXTH26) grouped into subfamily III-B, which may share similar functions to the other members of this family. *Rose et al. (2002)* associated four members (AtXTH1, AtXTH2, AtXTH3, and AtXTH11) of the *A. thaliana* XTH family in group 1 (now called ancestral), which present XET activities. Three SlXTHs (SlXTH30, SlXTH31, and SlXTH36) clustered into this family, whereas the rest of the SlXTH members grouped into the I/II subfamily, which includes most of the XTHs from *A. thaliana* predominantly exhibiting XET activity (*Rose et al., 2002*). All these results suggest that structural characteristics in the amino acid sequence of each XTH protein might result in a high possibility of functioning as XEH instead of XET and support the idea that XTH proteins might cluster according to their functional activity in different plants (*Song et al., 2018*; *Wang et al., 2018*; *Fu, Liu & Wu, 2019*; *Li et al., 2019*; *Wu et al., 2020*; *Zhu et al., 2022*; *Qiao et al., 2022*; *Yang, Zhang & Zhou, 2022*). Additional studies are needed to confirm whether each subfamily of the SlXTH protein family has XEH, XET, or combined functions.

Signal peptides are short sequences located in the N-terminal end of proteins that determine their entrance into the protein secretion pathway and target proteins to their final location in the cell. These signals play essential roles in cellular functions, such as cell proliferation and differentiation, transmembrane transport, and synthesizing new proteins involved in the cell wall expansion (*Owji et al., 2018*). Putative signal peptides are found in 34 out of the 37 SlXTHs, indicating that these proteins are transported to, and associated with, the plasma membrane. Consistently, almost all SlXTH proteins found *in silico* are located in the plasma membrane, except four SlXTHs situated in the extracellular region of the cell. These results agree with the XTH localizations reported in other monocotyledonous and dicotyledonous plants (*Song et al., 2018*; *Fu, Liu & Wu, 2019*).

Gene mapping positions demonstrated an uneven distribution of the 37 *SlXTH* genes in the 12 tomato chromosomes, which can be used to correlate the evolution of tomatoes with other plant species (*Wu & Tanksley, 2010*; *Fu, Liu & Wu, 2019*; *Wu et al., 2020*). In this work, four *SlXTHs* arranged in two homologous pairs (*SlXTH3/SlXTH37* and *SlXTH 24/SlXTH35*) were confirmed to be the result of tandem duplication events. On the other hand, 24 of the 37 *SlXTHs* were identified as having arisen as segmental events. This could have increased the functional divergence among XTH members, and suggests that duplication events were likely involved during plant evolution and that they have played roles in expanding multigene families in plant species (*Panchy, Lehti-Shiu & Shiu, 2016*; *Clark & Donoghue, 2018*), such as with the *SlXTH* gene family.

In tomatoes, duplications were estimated to occur approximately 7.4 million years ago by the Ka/Ks ratio. This divergence time is consistent with findings in *B. oleracea*, *N.*

*tabacum* and *S. superba*, where segmental duplication occurred 10 million years (*Wang et al., 2018*; *Wu et al., 2020*; *Yang, Zhang & Zhou, 2022*).

The correlation between duplication events and common *cis*-acting regulatory elements was previously reported (*Flagel & Wendel, 2009*; *Arsovski et al., 2015*; *Zhao et al., 2020*). Our study shows that most *SlXTH* gene duplicated pairs present common *cis*-acting regulatory elements in their promoter region. Regulatory elements play essential roles by modulating the transcriptional gene expression (*Zhu et al., 2022*). This study found various phytohormone and defense/stress-responsive elements in the promoter regions of *SlXTH* genes, including MeJa-responsive and W-box elements, essential factors regulating plant responses to abiotic and biotic stresses.

Regarding our *in silico* expression analysis using previously reported transcriptomic data, 19 *SlXTH* genes were expressed across all tissues examined from the databases. Interestingly, some *SlXTH* genes were found to be highly expressed in leaves (*SlXTH5*, *SlXTH7*, and *SlXTH16*), roots (*SlXTH14* and *SlXTH6*), flowers (*SlXTH1* and *SlXTH21*), and buds (*SlXTH14* and *SlXTH16*), suggesting that these genes may play an important role during cell differentiation in tomato. Differential expression patterns of *XTHs* have also been found in other plant species (*Wang et al., 2018*; *Wu et al., 2020*).

It is well known that AM symbiosis affects the expression profile of plant genes for the plant to accommodate the fungal symbiont in the roots and to adjust its responses to the symbiotic interaction, such as for improved nutrient and water acquisition and responses to abiotic and biotic stresses (*Miozzi et al., 2019*; *Sanmartín et al., 2020*; *Pozo de la Hoz et al., 2021*). In the present work, the expression profiles of nine *SlXTH* genes were evaluated in response to AM symbiosis. Even though we did not investigate the response of AM tomato tissues when challenged by a pathogen, it has already been reported that a priming mechanism is systemically induced by AM symbiosis that allows plants to improve their defenses against subsequent pathogen attack (*Pozo & Azcón-Aguilar, 2007*). We therefore hypothesize that mycorrhiza-responsive genes, such as some *XTH* genes, could be related to this defense priming mechanism. Although during early interaction between arbuscular mycorrhizal fungi and plant roots, some defense secondary metabolite accumulation occurs, the magnitude of this response is milder than the one observed during a pathogen attack (*Harrison & Dixon, 1993*). Similarly, while some defense genes are induced transcriptionally in mycorrhiza colonized roots, the expression profile of other genes differs and is less intense than in pathogen-infected tissues (*Pieterse et al., 2014*). Then, the plant can recognize AM fungus as a beneficial partner.

Our results reveal that AM symbiosis induces differential expression in most of the selected *SlXTH* genes in the leaves and roots of tomato plants. In leaves, seven *SlXTH* genes were downregulated in AM symbiotic plants, whereas one gene was upregulated (*SlXTH2*), and another was unaffected (*SlXTH17*). In the roots, however, only two *SlXTHs* (*SlXTH7* and *SlXTH35*) were upregulated, and two (*SlXTH3* and *SlXTH21*) were downregulated. Then, it cannot be ruled out that these differentially regulated *SlXTH* genes are involved in establishing symbiosis. However, additional studies must be done to confirm this possibility. Interestingly, *SlXTH17* was undetectable in tomato roots, indicating that this gene might be explicitly expressed in leaves. In a previous transcriptomic analysis in shoots

of AM-colonized *Medicago truncatula*, a putative *XTH* gene (MT001587) was also described to be upregulated (*Liu et al., 2007*), which is in agreement with the fact that *SlXTH2* was the only induced gene in AM tomato shoots. A multiple sequence alignment (ClustalW) comprised of MT001587 *M. truncatula* gene product (NCBI protein sequence accession RHN64771) with the 37 SlXTH protein family members described in the present work suggests that this gene could be orthologous to *SlXTH2*. The upregulation of these genes in shoots of AM-colonized plants in tomatoes and *M. truncatula* supports this idea. The fact that most *SlXTH* genes repress in leaves and only *SlXTH2* upregulates might indicate that XTH activity is highly regulated as a response in leaves to mycorrhiza colonization.

It can be hypothesized that both XTH enzymatic activities (endotransglucosylase and hydrolase) modify cell walls in fungal penetration of root cells during arbuscular mycorrhizal establishment. In tomatoes, we found at least four differentially regulated *SlXTH* genes in the roots of colonized plants. The coordination of the expression of these genes may intervene in the accommodation of the fungus within the hearts. In shoots, differentially expressed genes were also identified in colonized plants. In particular, *SlXTH2* was found to be upregulated, whereas seven *SlXTH* genes were downregulated. This suggests, in shoots, that cell wall modification might also occur in colonized plants. Furthermore, the rearrangement of the xyloglucan backbone in leaf cells of colonized plants by *XTHs* could strengthen their cell walls by making them less susceptible to subsequent pathogen attacks. Thus, in addition to facilitating fungal invasion in root tissues, modification of cell walls by *XTHs via* mycorrhiza colonization might also fortify shoot tissues to resist biotic stress better. These results are consistent with previous reports in other plant species, where several genes involved in cell wall biogenesis are upregulated in response to AM symbiosis (*Schoenherr et al., 2019*; *Sanmartín et al., 2020*; *Jiang et al., 2021*; *Pozo de la Hoz et al., 2021*). Specifically, the GO analysis and the predicted 3D protein structure indicate that these nine *SlXTHs* are involved in cell wall biogenesis by transferring and hydrolyzing xyloglucan in the cell wall. Finally, the results from this study will provide a foundation for further investigation of the function of *XTH* genes in tomato plants and their role in AM symbiosis.

## CONCLUSIONS

In this study, 37 *SlXTH* genes were identified and characterized in tomato (*S. lycopersicum*) using a comprehensive genome-wide analysis. All SlXTH proteins were classified into three subfamilies (ancestral subfamily, subfamily I/II, and subfamily III) by comparison with other XTHs from Solanaceae and *A. thaliana*. Structural genomic (exon/intron) and conserved motifs also support this classification. Evolutionary aspects in tomatoes revealed that the expansion of *SlXTH* genes occurs by tandem and segmental gene duplication. Through gene ontology (GO) annotation, we found that all SlXTHs participated in cell wall biogenesis and in xyloglucan metabolism, which is consistent with the function predicted by the 3D protein structure. The occurrence of certain *cis*-acting regulatory elements in the promoter region of *SlXTH* genes indicates their potential roles in cell development, defense and stress responses, and hormone signaling. Expression analysis in different

tissues revealed that some *SlXTH* members are differentially expressed in the leaves and roots of tomatoes in response to AM symbiosis. The such differential expression might be used to finely regulate the establishment of the fungus in root cells and strengthen leaf cells to reduce susceptibility to pathogens by rearranging cell wall components such as xyloglucans. Taken together, our research provides a comprehensive and systematic analysis of the *XTH* gene family in tomatoes and presents new sources for further investigations of the molecular role of *SlXTHs*.

## ACKNOWLEDGEMENTS

We thank Claudia María Ramírez-Douriet for her technical assistance.

### Funding

This work was supported by the Consejo Nacional de Ciencia y Tecnología (CB A1_S_31400), the Secretaría de Investigación y Posgrado-IPN (20196531, 20211500, 20222056); CONACYT for Luis G. Sarmiento-López and Maury Yanitze López-Espinoza's postdoctoral and master's scholarships, respectively; as well as BEIFI-IPN for Maury Yanitze López-Espinoza's scholarship. The funders had no role in study design, data collection and analysis, decision to publish, or preparation of the manuscript.

### Grant Disclosures

The following grant information was disclosed by the authors:
Consejo Nacional de Ciencia y Tecnología: CB A1_S_31400.
Secretaría de Investigación y Posgrado-IPN: 20196531, 20211500, 20222056.
CONACYT.
BEIFI-IPN.

### Competing Interests

The authors declare there are no competing interests.

### Author Contributions

- Luis G. Sarmiento-López conceived and designed the experiments, performed the experiments, analyzed the data, prepared figures and/or tables, authored or reviewed drafts of the article, and approved the final draft.
- Maury Yanitze López-Espinoza performed the experiments, authored or reviewed drafts of the article, and approved the final draft.
- Marco Adán Juárez-Verdayes performed the experiments, analyzed the data, prepared figures and/or tables, authored or reviewed drafts of the article, and approved the final draft.
- Melina López-Meyer conceived and designed the experiments, analyzed the data, authored or reviewed drafts of the article, and approved the final draft.

## Data Availability

The raw measurements are available in the Supplementary Files.

## Supplemental Information

Supplemental information for this article can be found online at http://dx.doi.org/10.7717/peerj.15257#supplemental-information.

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
