# Peer review of "Genome-wide characterization of the xyloglucan endotransglucosylase/hydrolase gene family in Solanum lycopersicum L. and gene expression analysis in response to arbuscular mycorrhizal symbiosis"

_PeerJ, doi:10.7717/peerj.15257_

## Round 0.1 · original submission · Minor Revisions

All three reviewers concur in requiring minor revisions to the manuscript. Please check all details and submit a revised version with a detailed rebuttal letter.

Reviewer 1 ·

Basic reporting

Here, the authors did a broad description of the XTH gene family in tomato using different genomic approaches and bioinformatic tools. Also, they evaluated the expression of some XTH genes in roots and leaves under symbiotic conditions with the mycorrhiza Rhizophagus irregularis. I consider that this is a work that increases the knowledge of the area and from which other important hypotheses can be derived.

Minor revisions:
-Lines 292-294: You are repeating information: “Structural analysis of SlXTH genes showed that the most closely related genes in each subfamily share similar exon and intron numbers, which was consistent with the phylogenetic analysis. For example, members from subfamily I/II contain similar intron and exon distributions.” You can specify the number of exon and intron in the second case.
-You have written some names of genes without italics, i.e. lines 30, 255, 264, 265, 268, 335, 338, 339, 341, 359, 360, 418, 419.
-Please indicate why you used sequences of tuberosum, N. tabacum, P. axillaris, and A. thaliana for the phylogenetic analysis (not only in the conclusion).
-In lines 455-457 you mention the duplication events, but you don’t discuss your results. I suggest discussing this here or passing the information below.
-In Fig. S1 I suggest that instead of showing the graph with the domain numbers, it’s too enriching to show a diagram where you locate the domains in all the proteins.
-In Fig. S2. You can organize the figure to show what proteins pertain to each family: I/II and III.


Major revisions:

All my major revisions are regarding the discussion section. Here, you make again a wide description of your results but you don't discuss in depth some important points. First, I suggest summarizing as far as possible the data and discussing it more. I have some points that could help you:

-You mention that the phylogenetic distribution could be associated with the function of proteins (lunes 471-478). What examples are about this in the literature? What is the function reported for each group of proteins (I/II, III) and how can you relate this with the phylogenetics?

-In Arabidopsis, if you mutate an XTH gen, the plants are more susceptible to pathogens. Despite you are working with mycorrhiza, there are some essential mechanisms in colonization shared by phytopathogens and symbionts. In this sense, why do you think you didn't find so many differentially expressed genes in the roots? Could you hypothesize that XTH3 and XTH21 could be playing an essential role in colonization? What about XTH7 and XTH35?. Even more, in leaves, what could be the reason why almost all genes are repressed? Do mycorrhizae also colonize the leaves? Please discuss all these in depth.

-If you don’t find information in tomatoes and mycorrhiza you could use the information reported for other plants and symbionts.

-Please review the following work: Shinohara, N., & Nishitani, K. (2021). Cryogenian origin and subsequent diversification of the plant cell-wall enzyme XTH family. Plant and Cell Physiology, 62(12), 1874-1889. I think that it could improve your discussion.

Experimental design

All the experiments are well described.

Validity of the findings

Please indicate the FDR that you used for your expression analysis derived from transcriptomic data.
The conclusion is well stated.

Additional comments

NA

Reviewer 2 ·

Basic reporting

The manuscript provides useful information to the audience. It is well written and structure with valid literature cited.

Experimental design

Valid experimental design provided.

Validity of the findings

Italicize the gene names throughout the document such as Lines 91-92. It is better to be consistent with the terminology, either say tomato XTH genes or SlXTH genes. No need to mention the complete scientific name throughout such as Lines 118-119.

Reviewer 3 ·

Basic reporting

To me all the four points here are addressed.

Experimental design

Also in this case all the four points here are covered.

Validity of the findings

Regarding the validity of the results obtained in qPCR analyses, as I wrote in the text I was wondering about the level of normalized relative gene expression that in most of teh cases is apparently too low. Please comment on that.

Moreover, I ask you to correctly name the SlXTH genes based on a common criteria based on phylogenetic relationships with XTH members form other species and also between S. lycoperisum memebrs.

Annotated reviews are not available for download in order to protect the identity of reviewers who chose to remain anonymous.

---

## Round 0.2 · Minor Revisions

Please consider the comments in the attached PDF regarding grammar and style, and submit a revised version at your convenience

---

## Round 0.3 · accepted · Accept

Thanks for the last revisions; your paper is now accepted in PeerJ.